# Leadership Styles of Rural Leaders in the Context of Sustainable Development Requirements: A Case Study of Commune Mayors in the Greater Poland Province, Poland

**Agnieszka Springer [1] , Katarzyna Walkowiak [2] and Arnold Bernaciak [3,*]**

[1]   Institute of Management, WSB University in Poznan, 61-895 Poznan, Poland;
    agnieszka.springer@wsb.poznan.pl

[2]   Faculty of Political Science and Journalism, Adam Mickiewicz University, 61-614 Poznan, Poland;
    katarzyna.walkowiak@amu.edu.pl

[3]   Institute of Economics and Finance, WSB University in Poznan, 61-895 Poznan, Poland

[*]   Correspondence: arnold.bernaciak@wsb.poznan.pl

**Abstract:** In the context of sustainable and durable development postulates, local leadership and rural governance is a matter of particular importance. As a local leader, the commune mayor should have a vision of the commune's development and possess the ability to influence the citizens and involve them in the process of realizing that vision. Such a manner of governance is the essence of transformational leadership, which, according to the multifactor model developed by Bass and Avoilo, facilitates the achievement of positive organizational and social results. The authors' research is an attempt to answer the question about the style of governance adopted by the political leaders of rural communes in Poland and its links with their engagement in social activation and the use of participatory tools. A survey was conducted on a sample of 49 commune mayors from the Greater Poland Province (43%) using the Multifactor Leadership Questionnaire (MLQ) and a questionnaire about public participation developed by the authors. The results indicate that components of transactional leadership predominate among commune mayors from the Greater Poland Province. Transformational leadership, which is largely based on the leader's charisma, is exhibited much less frequently. At the same time, the majority of commune mayors use only the most basic forms of participation, without going beyond the requirements specified in the provisions of the law. Those commune mayors who rely more strongly on transformational leadership more often use public participation techniques as well.

**Keywords:** local leadership; leadership styles; public participation; rural areas; sustainable development

## 1. Introduction

In Poland, rural areas are inhabited by over one-third of the population, which generates roughly a quarter of GDP. Since the political and economic transformation of 1989, both the level of development and the living conditions in rural areas have improved considerably. Even so, rural areas in Poland are much diversified throughout the country [1] and provide a different standard of living than cities. Among the key problems and challenges facing local authorities are demographic problems and the ageing society, a lack of innovations in rural areas, the much-needed diversification of activities [2,3], or a low quality of social capital, among other things. On the other hand, local communities in rural areas have a certain unique potential that, with the use of appropriate, effective policy covering social,

economic, and environmental aspects, can be used by the local authorities to promote local sustainable development. Effective leadership, the foundation of which is the ability to build good relations with stakeholders (inhabitants, entrepreneurs, non-governmental organisations, etc.) and the use of social potential, is considered to be an essential factor that affects the functioning of local communities and their success.

In local government units (including rural communes), people have many legal and institutional solutions at their disposal that serve to involve them in the decision-making processes on the local level. These include public consultations, petitions, public hearings, debates, citizens' initiatives, local referendums, and participatory budgeting. Most often, local authorities invite citizens, local action groups, and other stakeholders in the processes of long-term development planning, public space management, village renewal planning, and the creation of programs for solving social problems, etc. Rural areas have particularly deep-rooted traditions of direct democracy and civic gatherings as their most common forms. A gathering of village citizens is not only an opportunity to elect representative bodies, but also a meeting during which the people are able to report their problems and needs, express their vision for the development of the village, and work on village renewal projects. Although the subjects of this study are local leaders in a single region of Poland, the issues discussed can be viewed from a broader perspective as well, because the local community is the basic unit of social and economic development in many cultures and socio-political systems. In any culture and system, co-participation in the decision-making processes should be treated by its members as a duty and a privilege, and the role of the leaders should be to make such participation possible (see, e.g., [4–8]). This becomes especially important in the context of the challenges of sustainable development. This fact has been noted in the sphere of international policy and diplomacy—ensuring that citizens have a say in the governance processes is one of the key postulates of sustainable development, which is explicitly expressed in the 2030 Agenda for Sustainable Development. The sixteenth goal of this document, entitled "Peace, Justice, and Strong Institutions", envisions ensuring "responsive, inclusive, participatory, and representative decision-making at all levels" [9].

Meanwhile, in Poland, we see minimal involvement of citizens in different public initiatives [10,11]. Research confirms that, despite the existing formal framework, local communities have no sense of empowerment and actual influence on the decision-making process [11,12]. One reason for this situation can be found in the manner of governance. A commune mayor, as the local leader, should have the ability to influence the citizens in the pursuit of specific goals, "have a sense of cooperation with the group and of interdependence on others in the realization of these goals" [13]. Such a manner of governance is the essence of transformational leadership, which, according to the multifactor model developed by Bass and Avoilo [14] (1994), leads to the achievement of numerous positive organizational and social results.

The question therefore arises as to what style of governance Greater Poland commune mayors represent and whether it translates into public participation in the decision-making process. Determining the answer to this question would help to identify the most important challenges facing local political leaders and communities in rural areas, especially in the context of sustainable development.

The above question, although in this case it specifically concerns the local level, has a universal dimension as well. It is a question about the relationship between the style of leadership and the willingness to introduce changes, in particular those that would allow leaders to meet more requirements of durable and sustainable development. It is a question about the role and responsibility of leaders in this context, and about their willingness to consciously choose a style of governance that would facilitate the process of developing the socioeconomic system in a desirable direction. One of the most important tools local leaders could be widely using is citizen participation in the decision-making processes. The universal nature of this issue is demonstrated by the inclusion of the postulate of broad citizen participation in the decision-making processes in the 2030 Agenda for Sustainable Development.

## 2. Literature Review

Previous research covering Polish rural areas takes on the issues of the role and significance of these areas in pursuing the goals of sustainable development and reducing their pressure on the environment [15,16]. Zarożniak [16] (2016] presents the rural community as the agent of development, one of the most important goals of which is the improvement of the quality of life. Sometimes the postulates of sustainable and durable development require the adjustment of previous paradigms and directions of development. However, in order for these processes to proceed effectively, rural communes need capable leaders who will involve the local community in these activities. Pawlewicz and Pawlewicz [17] (2010) emphasize that public participation is especially important for the sustainable development of rural areas. They point out that it is precisely the local community who can best define the main problem areas, threats, and development opportunities, and without its participation in the decision-making process all development strategies can turn out to be inadequate.

Research into political leadership and its role in social and economic transformation is well-described in the literature. These issues have been taken up by numerous authors (see, e.g., [18–22]). Previous research focuses on different aspects of the subject matter. An analysis of the specificities of administration and leadership in cities of different sizes was undertaken by Greasley and Stoker [23] (2009) and Savith and Tharp [24] (1997), as well as Bowers and Rich [25] (2000). The effect of leadership and optimization of governance processes on the quality and scope of provided public services was the subject of research conducted by Teelken, Ferlie, and Dent [26] (2012), as well as Milner and Joyce [27] (2005). In turn, Silva and Bucek [28] (2017), Michałowski [29] (2008), and Back [30] (2005) are among the researchers who took up the issue of the role of leadership in involving the public in governance processes. The role of local political leadership in social and economic transformation in the countries of Central and Eastern Europe has also been the subject of many studies, among them those of Borraz and John [31] (2004), Kersting and Vetter [32] (2003), and John [33] (2001).

A separate issue discussed in the literature concerns leadership in rural areas. The significance of transformational leadership for the development of rural areas is pointed out by Ricketts [34] (2005). Based on the results of research conducted in North Florida, she concludes that, in rural communities, it is important to define the vision and paths of development; two other vital elements in the development process are building trust and providing support, which is why she believes in the crucial role of transformational leaders who are able to rise to these challenges and mobilize local communities [34,35]. Davies [36] (2009) also states that an important factor in the long-term social and economic stability of special units is local leadership. Good leadership mobilizes existing resources and attracts new ones. Based on research conducted in six Australian rural communities, it was found that transformational leadership is more effective in this respect than a more transactional model [37]. The author emphasizes that, although transactional leadership can play an important role in shaping rural communities (e.g., due to its effectiveness in project management), more attention should be paid to supporting transformational leadership, as it is more effective in improving the adaptation abilities of society. The results of this research go hand in hand with the multifactor leadership model developed by Bass and Avolio. Bass [38] (1999) argues that these two leadership styles should be perceived as complementary and not as two extremes of the same continuum. The leadership model expanded and modified by Bass [38] (1999) includes nine aspects, five of which characterize transformational leadership, and four characterize transactional leadership [39]. Although a leader often employs both leadership styles, transformational leadership is more satisfying and effective than transactional leadership based on contingent rewards, which, in turn, is more satisfying and effective than expressing expectations and laissez-faire leadership [40]. The concept of multifactor leadership was adopted as a theoretical framework for the research, which, in this respect, was conducted only fragmentarily in Poland.

The research on leadership in Polish local governments is focused mainly on the circumstances and role of political leaders in the processes of social and economic development [29,41–46]. The first researchers to attempt to identify the governance styles of local leaders in the Polish formal, legal,

organizational, and cultural environment were Swianiewicz and Klimska [47] (2003). The subjects of their research were the mayors of two cities in the Greater Poland Province: Poznań and Ostrów Wielkopolski. An analysis of the governance styles adopted by mayors of Greater Poland cities was done by Springer, Bernaciak, and Walkowiak [48] (2018). As far as Polish rural areas are concerned, researchers have mainly focused on the circumstances of the functioning of local governments and citizen participation in governance. The study by Mazur and his co-authors [49] (2018) was aimed at identifying the specificity and temporal and spatial variability of the circumstances of the functioning of local governments in rural areas. It showed that the creation of a local leader is mainly determined by their personality traits and the development program they offer, and is largely unrelated to the locational, historical, social, or demographic reality of the given spatial unit [49].

The role of local leaders in the transformation of rural areas in Malaysia is emphasized by Rami, Abdullah, and Simin [50] (2017). Based on the results of studies conducted in that country, the authors conclude that the effective bridging of the social and economic gap between rural and urban areas was possible thanks to the commitment to and adoption of the correct governance style by the leaders. The key to their success was the fact that they were able to properly influence citizens in order to involve them in the transformation process. For this reason, we asked ourselves what the level of citizen participation in the studied communes is and if there is a link between the level of citizen participation and the favored governance style of the commune's mayor. Commitment and support for the local community, as well as building an atmosphere of mutual trust by the local leader, promote the identification and implementation of new sustainable development paths, as argued by Horlings and Padt [51] (2011) based on studies conducted in rural areas in the Netherlands. The basis for mobilizing private and public entities for the implementation of development goals are elements such as shared values, feelings, energy, trust, and commitment.

The study by Marks-Krzyszkowska and Michalska-Żyła [52] (2018) conducted in rural areas in Łódź Province showed the moderate willingness of citizens to influence the decisions of local authorities. The higher this willingness is, the higher are the levels of trust in local authorities and satisfaction with the way the commune is governed and with the influence citizens have on important communal issues, as well as their level of knowledge about the activities of the local authorities. Kulig, Miśkowiec, and Ogórek [53] (2018), in a case study of Olkusz commune, show the significance of public participation in the processes of revitalizing rural areas. In this case, the real commitment of the rural community made it possible to identify the potential and needs connected with revitalizing the areas studied.

As noted by Kosmaczewska [54] (2009), the willingness of society to participate in the process of governance and to solve the problems of the local community depends mainly on the quality of social capital. One of the factors promoting the development of social capital is the activity of local leaders [55]. Chodkowska-Miszczuk, Biegańska, and Grzelak-Kostulska [56] (2017) present the example of Jeżowo commune and the improvement of its social capital as a result of the commitment of its local leader, who managed to increase the level of interest in the activities she had initiated, institutional support, and effectiveness in acquiring funding.

Citizen participation hinges on the scope and quality of the applied mechanisms of involving the public in the decision-making process. The study conducted by Inglot-Brzęk [57] (2017) indicates that the majority of Polish local leaders apply tools related to providing information, asking for opinions, and conducting consultations, but they very rarely introduce the tools of co-decision.

Based on the knowledge of local leaders' governance styles, the following hypothesis was formulated:

**Hypothesis 1 (H1):** *Transactional leadership components predominate among commune mayors in the Greater Poland Province.*

Based on the tenets of the multifactor leadership model, which indicate the existence of a relationship between the applied governance styles and the subsequent results, another hypothesis was formulated:

**Hypothesis 2 (H2):** *The transformational leadership components used by commune mayors are more strongly correlated with positive results (such as satisfaction, extra effort, and effectiveness) than the transactional leadership components.*

At the same time, despite the fact that citizens express a willingness to participate in the public sphere, the leaders of rural communes most often will inform the public about their decisions or ask their opinions but rarely introduce a higher level of participation, i.e., co-decision. Another hypothesis was therefore formulated:

**Hypothesis 3 (H3):** *Most commune mayors apply only the basic forms of participation, without going beyond the legal requirements.*

Keeping in mind that transformational leadership is connected with establishing a vision for development and considering individual needs, another hypothesis was formulated:

**Hypothesis 4 (H4):** *Commune mayors who more often apply transformational leadership components in their governance, more often use citizen participation techniques as well.*

To sum up the previous research, it is worth highlighting the role played by local leaders in developing social capital, activating endo- and exogenous development factors of local social and economic systems, and establishing the directions of their transformation. This becomes especially important in the context of sustainable and durable development postulates: redefinition of previous development patterns and broad social involvement.

In light of previous scientific findings, it is not possible to indicate transformational governance as a factor that unambiguously contributes to the realization of the postulates of sustainable development. However, these findings allow us to identify the potential of the particular elements of this style of governance to include local communities in development processes (including decision-making processes), increase their social capital, create an atmosphere of trust between their members, and attract new resources. Sustainable development, seen as a model pursued by the local community, requires a change of commonly accepted paradigms, beliefs, and models. This, in turn, requires a change in awareness of its members. Thanks to individualized treatment, a transactional leader knows the needs of the members of his community and can make sure they have appropriate conditions for learning, acquiring new competences, and transcending previous limitations. Thanks to intellectual stimulation, he is able to free up innovativeness and creativity. He is able to inflect others with his vision of the future by consistently pursuing its realization and skillfully involving the members of his community in the process. The use of the elements of transactional governance by local leaders is not a necessary condition of implementing the postulates of sustainable development, but it can make it much easier to introduce changes that serve to increase the durability of local socioeconomic systems. Such an approach seems to be favored by the authors of the 2030 Sustainable Development Strategy, who demand that leaders at all levels ensure broad citizen participation in the decision-making processes. However, no elements of transactional governance can be identified as facilitating the realization of the postulates of sustainable development. In this regard, they should rather be perceived as neutral—when used by the leader, they can support actions initiated thanks to the use of the elements of transformational governance.

The openness of political leaders in rural areas and their adoption of the transformational approach to governance is an important factor in speeding up the process of transforming local social and economic systems, making them more durable, and helping them to develop more sustainably. So far,

no effort has been made to identify the governance styles of Polish local leaders in rural areas and their effect on durability and sustainable development. This study therefore provides new knowledge about governance in rural areas, filling the cognitive gap by answering questions about the governance styles of Polish local leaders in rural areas and their correlation with engagement in social activation and the use of participatory tools.

## 3. Materials and Methods

The research sample was composed of active commune mayors in the Greater Poland Province. A request to participate in the survey and fill out the questionnaires was sent in the second half of 2018 to the official email addresses of all rural communal offices in the Greater Poland Province (N = 113). The emails included the main message and a link to the questionnaire, and they were followed up by phone conversations. Eventually, responses from 49 commune mayors were received (43% of the sample). The majority of the respondents were male (there were only six women in the sample). The average age of the respondents was 55, and the average period of time they had held their position was 13 years. The breakdown of respondents by sex reflected the breakdown of the studied population—88% of the respondents were male and 12% were female (among commune mayors, almost 10% are women).

The research tool consisted of two parts; the first part diagnosed the style of governance and its effects, and the second included questions about the methods of citizen participation. In the first part, the authors used a shortened version (45 questions) of the Multifactor Leadership Questionnaire (MLQ) developed by Avolio and Bass. A self-report version was used, i.e., the commune mayors assessed the frequency of certain behaviors on a scale from 0 (never) to 4 (very often). Some questions were rescaled, and then the average values for nine measures characterizing governance styles and three measures of results were calculated. Eventually, each respondent was characterized according to:

- Four measures of transactional leadership (contingent rewards, active management by expectation, passive management by expectation, laissez-faire);
- Five measures of transformational leadership (idealized influence (attribute), idealized influence (behavior), inspirational motivation, intellectual stimulation, individualized consideration);
- Three measures of results (employee satisfaction, effectiveness, extra effort).

All measured scales, including the study results, constituted an integral part of the MLQ and were based on the subjective assessment of the leaders, e.g., "I lead a group that is effective". In the case of all scales, the results were compared with the normalized sample prepared for the version based on self-assessment. For this reason, although it is not impossible that the self-assessment results were higher than the results based on the assessment by others, when identifying the levels (low, medium, high), we followed the distributions of studies conducted in this form.

The second part of the questionnaire included questions developed by the authors in order to assess the level of the respondents' openness to citizen participation. The respondents were asked about the frequency of using the consulting techniques listed in the questionnaire on a scale from 1 (never or very rarely) to 5 (very often). The questionnaire listed 11 techniques: meetings with representatives of support units, with citizens, with non-governmental organizations, with experts, and with businesses; participation of non-governmental organizations or citizens in committee meetings or city/commune council sessions; collecting opinions in writing, via the Internet, councilmen, or permanent consultative and advisory councils; and deliberative polling. Additionally, the respondents were asked if the public consultations in their communes go beyond the legal requirements, which allowed the researchers to determine whether local authorities want to open up to the local community more than they need to.

## 4. Results

In line with the aim of the study, the authors first identified the commune mayors' governance styles and the relationship between the use of transactional and transformational leadership components and organizational results (see Table 1).

**Table 1.** Multifactor Leadership Questionnaire (MLQ) results: average, standard deviation, and Pearson's r.

| | M | SD | Transformational Leadership | | | | | Transactional Leadership | | | | Results | |
|---|---|---|---|---|---|---|---|---|---|---|---|---|---|
| | | | IIA | IIB | IM | IS | IC | CR | MBEA | MBEP | LF | EE | EFF |
| IIA | 2.73 | 0.58 | 1 | | | | | | | | | | |
| IIB | 3.17 | 0.53 | 0.476** | 1 | | | | | | | | | |
| IM | 2.98 | 0.50 | 0.631** | 0.604** | 1 | | | | | | | | |
| IS | 3.07 | 0.50 | 0.280 | 0.732** | 0.491** | 1 | | | | | | | |
| IC | 2.93 | 0.51 | 0.215 | 0.678** | 0.422** | 0.714** | 1 | | | | | | |
| CR | 3.23 | 0.51 | 0.347* | 0.804** | 0.527** | 0.706** | 0.720** | 1 | | | | | |
| MBEA | 2.96 | 0.63 | 0.376** | 0.618** | 0.496** | 0.630** | 0.468** | 0.509* | 1 | | | | |
| MBEP | 1.51 | 0.72 | 0.121 | −0.183 | 0.066 | −0.198 | 0.009 | −0.309* | −0.112 | 1 | | | |
| LF | 0.81 | 0.67 | −0.207 | −0.462** | −0.267 | −0.397** | −0.290* | −0.559** | −0.366** | 0.560** | 1 | | |
| EE | 2.87 | 0.54 | 0.451** | 0.672** | 0.536** | 0.606** | 0.633** | 0.542** | 0.482** | 0.005 | −0.298* | 1 | |
| EFF | 3.09 | 0.52 | 0.476** | 0.726** | 0.504** | 0.733** | 0.630** | 0.712** | 0.449** | −0.220 | −0494** | 0.776** | 1 |
| SAT | 2.95 | 0.58 | 0.320* | 0.455** | 0.430** | 0.433** | 0.536** | 0.512** | 0.232 | −0.032 | −0294* | 0.552** | 0.615** |

Notes: * p value = 0.05; ** p value = 0.01; IIA—Idealized Influence (Attribute); IIB—Idealized Influence (Behavior); IM—Inspirational Motivation; IS—Intellectual Stimulation; IC—Individualized Consideration; CR—Contingent Reward; MBEA—Management-by-Expectation (Active); MBEP—Management-by-Expectation (Passive); LF—Laissez-Faire; EE—Extra Effort; EFF—Effectiveness; SAT—Satisfaction.

Table 1 presents the averages for the obtained scales and the strength of the relationships between them. The obtained averages for the leadership scales were used to identify the style of governance in the studied group, and the correlation coefficients allow the researchers to preliminarily assess the relationship between a given component and the study results; they were subsequently introduced into the regression models described below.

The raw average result on the scale of leadership components is not enough to be able to assess the level of use of the particular leadership components by the commune mayors in the Greater Poland Province. To make such an assessment possible for all scales studied, each commune mayor was assigned a level—low, medium, or high. This categorization was made based on a comparison of the obtained result with the distribution of percentile results illustrating the distribution of results obtained in the normalized sample, which is included in the MLQ manual.

First, the authors analyzed behaviors that are characteristic of transactional leadership, which, on the one hand, is necessary to achieve results but, on the other hand, can make it more difficult to realize organizational goals. The first group includes contingent rewards and active management by expectation. The results show that the commune mayors govern by setting goals—96% (see Figure 1) of them obtained results categorized as high. At the same time, the high results for passive and avoidance behaviors, which can be indicative of the commune mayors' belief that there is no need for them to be more engaged in the governance process, are worrisome.

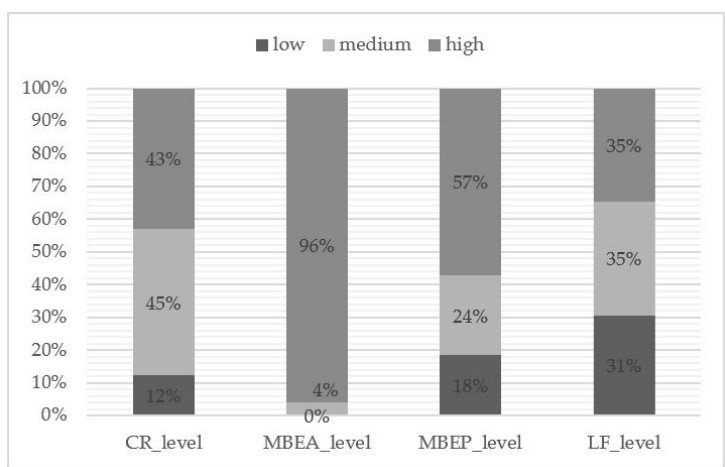

**Figure 1.** Distribution of the frequency of use of particular transactional leadership techniques. Notes: CR—Contingent Reward; MBEA—Management-by-Expectation (Active); MBEP—Management-by-Expectation (Passive); LF—Laissez-Faire.

At the same time, the results show that the frequency of using transformational leadership techniques varies significantly; the least used techniques are Idealized Influence connected with Attributes and Individualized Consideration (see Figure 2). It is worth noting, however, that almost half of the respondents engage in intellectual stimulation, which is indicative of their openness to the knowledge and ideas of other members of the organization.

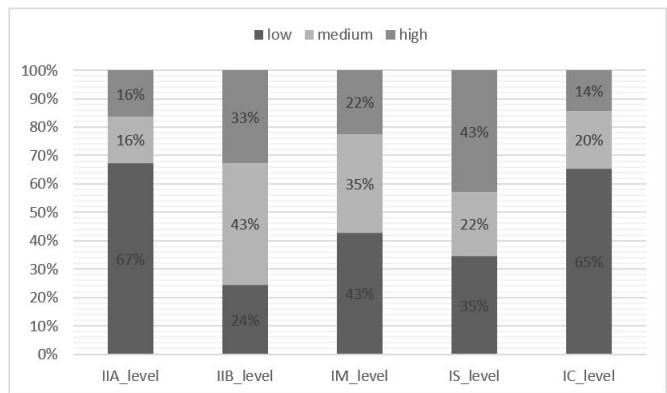

**Figure 2.** Distribution of the frequency of use of particular transformational leadership techniques. Notes: IIA—Idealized Influence (Attribute); IIB—Idealized Influence (Behavior); IM—Inspirational Motivation; IS—Intellectual Stimulation; IC—Individualized Consideration.

The above results support the first research hypothesis, which says that transactional leadership components are predominant when it comes to the governance style of Greater Poland commune mayors.

From the point of view of management practice, it is not only important to determine the frequency of using techniques of different governance styles, but also the strength of their correlation with the achievement of positive results. In the case of all measures of transformational leadership and two constructive measures of transactional leadership, we can see a statistically significant (moderate or high) correlation with expected results such as satisfaction, extra effort, and effectiveness. In the case of the laissez-faire model, this correlation is statistically significant and inversely proportional, while passive management by expectation does not show any statistically significant correlation.

Taking the studied dimensions of leadership as independent variables and the expected results as dependent variables, three regression models were calculated. The analysis showed that, for both employee satisfaction and extra effort, the key variable is individualized consideration and, additionally, for extra effort, idealized influence (behavior). Employee effectiveness is affected by at least two dimensions of transformational leadership; intellectual stimulation and idealized influence (attribute), and one dimension of transactional leadership; contingent rewards (see Table 2). The results support the second hypothesis pertaining to a larger role being played by transformational leadership in the achievement of positive results.

**Table 2.** Regression equation parameters.

| Dependent Variable and Model Parameters | Independent Variables | B | Standard Error | *p*-Value |
| --- | --- | --- | --- | --- |
| SAT R-squared = 0.28; F = 18.94; p = 0.000 | (Constant) | 1.157 | 0.418 | 0.008 |
| | IC | 0.612 | 0.141 | 0.000 |
| EE R-squared = 0.51; F = 23.9; p = 0.000 | (Constant) | 0.375 | 0.365 | 0.310 |
| | IIB | 0.463 | 0.145 | 0.002 |
| | IC | 0.351 | 0.150 | 0.024 |
| EFF R-squared = 0.664; F = 2.968; p = 0.000 | (Constant) | 0.061 | 0.326 | 0.852 |
| | IS | 0.454 | 0.125 | 0.001 |
| | IIA | 0.215 | 0.082 | 0.011 |
| | CR | 0.323 | 0.128 | 0.015 |

Another aspect of the study concerned involving the stakeholders in the process of communal management. According to the rural leaders, the most frequently used techniques include meetings with citizens; deliberative polling is used less often (see Table 3). It is worth noting, however, that although their declarations indicate a high level of participation, at the same time, 60% of them state that these activities do not go beyond the legal requirements, which supports the third hypothesis.

**Table 3.** Frequency of using particular techniques of citizen participation*.

| High Frequency | Moderate Frequency | Low Frequency |
|---|---|---|
| Meetings with citizens (88%) | Meetings with non-governmental organizations (50%) | Collecting written opinions (23%) |
| Meetings with representatives of support units (84%) | Participation of non-governmental organizations or citizens in committee meetings or city/commune council sessions (50%) | Collecting opinions from businesses (23%) |
| Collecting opinions via councilmen (70%) | Collecting opinions via the Internet (40%) | Collecting opinions via permanent consultative and advisory councils representing stakeholder groups (19%) |
| | Meetings with experts (39%) | Deliberative polling (15%) |

Notes: * The numbers in parentheses represent the proportion of commune mayors who frequently or very frequently use a given participation technique.

The last question this study attempted to find an answer to concerned the relationship between the particular leadership components and the use of citizen participation techniques. An analysis of the correlation relationships between the results of the MLQ scales and the sum of points obtained in the questionnaire about the frequency of using particular participation techniques was therefore performed. Moreover, the differences in MLQ scales between commune mayors who do not go beyond the legal requirements concerning participation and those who do were compared. In the first case, a statistically significant positive correlation was found between the use of citizen participation techniques and four out of six transformational leadership components and two out of four transactional leadership components (see Table 4). The local leaders were rather eager to declare that they use a variety of participation techniques, but at the same time only 40% of them said that they go beyond the legally required minimum, which is why the authors decided that the latter information could be crucial for the identification of the relationship between openness to citizen participation and the chosen style of governance. In the distinguished groups, a test of the significance of differences in the obtained results on the leadership scales was performed. Statistically significant differences were found regarding the higher results on five out of six transformational leadership scales and on the transactional leadership contingent reward scale (see Table 5) obtained by those commune mayors who go beyond the legal requirements in their use of participation techniques. These results support the fourth hypothesis concerning the relationship between transformational leadership and engagement in citizen participation activities.

**Table 4.** Person's r between leadership dimensions and the frequency of using participation techniques.

| | IIA | IIB | IM | IS | IC | CR | MBEA | MBEP | LF |
|---|---|---|---|---|---|---|---|---|---|
| Use of participation techniques*** | 0.260 | 0.401** | 0.237 | 0.436** | 0.383** | 0.339* | 0.352* | 0.159 | −0.033 |

Notes: * p value = 0.05; ** p value = 0.01; *** the coefficient was calculated as a sum of points for the frequency of using the 11 listed techniques of citizen participation.

**Table 5.** Relevance of differences on MLQ scales between commune mayors who use participation techniques more and less frequently, respectively.

|  | Meets Minimum | Exceeds Minimum | *p*-Value |
|---|---|---|---|
| IIA | 2.6724 | 2.8125 | 0.433 |
| IIB | 3.0086 | 3.4125 | 0.009 |
| IM | 2.8621 | 3.1500 | 0.037 |
| IS | 2.9397 | 3.2625 | 0.019 |
| IC | 2.7931 | 3.1250 | 0.021 |
| CR | 3.0862 | 3.4500 | 0.007 |
| MBEA | 2.8534 | 3.1250 | 0.147 |
| MBEP | 1.4655 | 1.5833 | 0.579 |
| LF | 0.8190 | 0.8000 | 0.926 |

Notes: IIA—Idealized Influence (Attribute); IIB—Idealized Influence (Behavior); IM—Inspirational Motivation; IS—Intellectual Stimulation; IC—Individualized Consideration; CR—Contingent Reward; MBEA—Management-by-Expectation (Active); MBEP—Management-by-Expectation (Passive); LF—Laissez-Faire.

## 5. Discussion

Transactional leadership components predominate in the governance styles of leaders in the rural areas of the Greater Poland Province. It is worth noting that the profile of components used in this group does not differ from the one obtained in the group of city mayors [48], which can be indicative of the fact that the office governance style depends on the rural or urban character of the commune. What is more, a similar governance style was observed among the leaders of Polish enterprises [58]. Thus, irrespective of the type of organization, there is a homogenous approach to human resource management, which is focused primarily on the realization of tasks; this suggests that situational factors are of secondary importance, and the competences (or the lack thereof) and habits of the leaders are crucial. On the one hand, the predominance of transactional leadership components contributes to their perception as strong leaders who are prepared to implement the necessary changes; on the other, it requires us to ask about the directions of these changes and whether they fulfil the needs of the local community and the demands of the modern world, including the postulates of sustainable and durable development in particular.

For the majority of the leaders surveyed, expressing the expectations of employees is a fundamental tool of governance. Competent task setting is an important attribute of a leader and an ability that is indispensable to the efficient management of development processes in rural areas. In the studied sample, however, it often takes a passive form—it is not accompanied by information about how these tasks are to be or can be achieved, which can have a negative effect on the likelihood of their achievement.

The predominance of transactional leadership and the relatively small share of transformational leadership components have a negative impact on the possibility of building a civil society and ensuring sustainable and durable development. Rural communities require their leaders to paint a clear vision—to define the paths that are to lead them to a better quality of life. This is especially important when it comes to a redefinition of the previous development paradigms and a turn towards the postulates of sustainable and durable development [16]. It seems paramount in these circumstances to build collective trust among citizens. A sense of support from leaders is also not without significance [34]. As a result, we could see an improvement of the adaptation abilities of society [36]. However, these behaviors are characteristic of transformational leadership, the elements of which are not commonly used by commune mayors in the Greater Poland Province. In view of the previous findings, according to which elements of transformational governance promote increased durability and sustainability of development, and elements of transactional governance are rather neutral in this respect, it needs to be stated that this limits the possibilities of implementing the postulates and realizing the goals of sustainable development.

The results of the study allow us to state that the leaders of the spatial units studied influence the behavior of their employees. They use Intellectual Stimulation (IS)—setting tasks that require them to find creative solutions relatively frequently. At the same time, they have a relatively minor influence on their employees' attitudes (Idealized Influence (Attribute) (IIA)). Individualized Consideration (IC) is also rare. As in many other studies [59–61], in the case of commune mayors, it was also found that there is a correlation between transformational leadership components and employee satisfaction and engagement. Moreover, in line with our assumptions, transformational leadership and Contingent Rewards (CR) turned out to be a predictor of taking external actions addressed to the local community. Local leaders define development goals and tasks, but they do not take measures to build a local community, strengthen the sense of identity, or increase trust. This is also indicated by the low level of openness to citizen participation. Commune mayors stick to the basic participatory tools they are required to use by law. This does not promote the participation of local communities in defining development goals, as postulated by Pawlewicz and Pawlewicz [17] (2010). It is not compliant with the postulate of ensuring responsive, inclusive, participatory, and representative decision making, as specified in the 2030 Agenda for Sustainable Development [9].

There exist significant differences in the scope of the use of participatory tools between leaders of local communities in rural and urban areas. In cities, participatory tools are used that go beyond the legal requirements: consultative and advisory councils, meetings with non-governmental organisations, businesses, and experts [48]. Commune mayors do not attach much importance to these tools.

## 6. Conclusions

The need to transform the socioeconomic structures and put them on the path of sustainable development is a global challenge, but the real actions in this regard are taken on the local level. This truth is best captured in the well-known saying connected with the concept of sustainable development: think globally, act locally. The efficiency of local actions largely depends on local leaders. The right combination of components of different governance styles can be an important factor in changes that increase the sustainability and durability of social and economic development of rural areas. A suitable share of transformational governance components is necessary. The study showed that commune mayors in the Greater Poland Province rarely use these elements. Transactional components predominate in their governance. Commune mayors serve the function of heads of office and team managers more than leaders of local communities. This limits their possibilities of implementing the postulates of sustainable and durable development. The differences in this regard between rural and urban areas make it necessary to organize support initiatives (educational, organizational). Participatory education seems especially important in the context of emigration and remigration from the cities to the countryside of citizens interested in more innovative forms of participation in planning and implementing local policy.

Involving local communities in the decision-making processes could increase their adaptation abilities and, as a result, mobilize existing and attract new resources [36].

The conducted study was spatially limited, and its results can only be directly applied to the studied area (Greater Poland Province). However, the universal character of the issues discussed (leadership, participation) allows for a slightly broader interpretation of the results. As noted before, the local community, irrespective of the culture or socio-political system, is the basic unit of socioeconomic development. It is where added value is created, where innovations are made, where leaders grow up, and where the process of change is initiated. The functioning of a local community largely depends on its leaders and their chosen style of governance. As showed by previous studies conducted in other parts of the world [34,36,50,51] and by the findings presented in this paper, an appropriate composition of governance elements, with a larger share of transformational than transactional elements, is of great importance. Transformational governance is more effective at opening the leaders up to the local community. The significance of the changes related to the socioeconomic transformation in the direction of sustainable development is so great that it should be realized with the widest possible

degree of citizen participation. This is why the very important postulate of citizen participation found its way into the sixteenth goal of the Agenda for Sustainable Development.

The identification of the shortcomings in the use of the elements of transformational governance by the commune mayors who are active in the Greater Poland Province has a cognitive value. It also allows to word practical postulates addressed to the local leaders about increasing the share of transformational governance elements in their current activity and being open to citizen participation in the decision-making processes. In view of the previous studies discussed above, as well as the citizen participation objectives included in the Agenda for Sustainable Development, a document with global reach, these postulates can be considered universal. Increasing the share of transformational governance elements in the activity of local leaders and their openness to citizen participation in the decision-making processes are relevant postulates irrespective of geographic location, socio-political system, and, to a large extent, of cultural reality as well.

Continuing these studies so as to cover the whole country seems warranted. It would also be interesting to conduct comparative studies among local leaders in different European and non-European regions. Such studies could more broadly verify the actual role of local leaders in the processes of socioeconomic transformation, identify good practices, and indicate shortcomings in this regard.

The conducted studies can serve as a starting point for a further analysis of the changes in contemporary leadership in local government institutions, especially in terms of local development management with citizen participation. These novel studies are also a contribution to the analysis of public administration management, because our knowledge of local government leadership comes primarily from studies conducted in private-sector institutions—so far, analyses of local governance styles have been rare. From this perspective, the results of the studies allow us to draw some important conclusions and make recommendations for the local leaders, supporting them in terms of the further development of suitable leadership behavior. The conclusions drawn from this paper will also be useful for theoreticians and social activists who wish to increase the effectiveness of the actions they take in their local communities.

**Author Contributions:** Conceptualization, A.S., K.W., and A.B.; methodology, A.S., K.W., and A.B.; software, A.S.; validation, A.S., K.W., and A.B.; formal analysis, A.S.; investigation, A.B., A.S., and K.W.; resources, K.W. and A.B.; data curation, A.S.; writing—original draft preparation, A.S., K.W., and A.B.; writing—review and editing, A.S., K.W., and A.B.; visualization, A.S.; supervision, A.S., K.W., and A.B..; project administration, A.B.; All authors have read and agreed to the published version of the manuscript.

**Funding:** This research received no external funding.

**Conflicts of Interest:** The authors declare no conflict of interest.

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
