# Peer review of "Leadership Styles of Rural Leaders in the Context of Sustainable Development Requirements: A Case Study of Commune Mayors in the Greater Poland Province, Poland"

_sustainability, doi:10.3390/su12072676_

Round 1

Reviewer 1 Report

In general I thought this was a solid paper. When reading the abstract, I was immediately wondering if the paper belonged in Sustainability. However, although it could be further strengthened, the authors do a reasonable job of establishing the relevance to sustainable development. The paper seems to have potential for publication. Please see my specific comments below.

  • The authors do a good job of locating this research in Poland. However, an article written for an international journal such as sustainability MUST establish the importance and relevance of the key issues (leadership, public participation) not only in Poland but also more broadly. This should appear more prominently in Section 1  and then in the final sections.
  • H-2 states: "The transformational leadership components used by commune mayors are more strongly 180 correlated with positive results than the transactional leadership components." In a hypothesis, you must be more specific in the delineation of the dependent variable than "positive results".
  • Section 3: How did the sampled participants differ by gender, age and experience from the population?
  • Although your results do not suggest a single source bias, I am always wary about relying leaders to self-rate. Please address this issue for the MLQ (self v. other ratings) and why it is not a problem in this study.
  • I generally found it awkward to read your Figures and tables due to the abbreviations. Figure 1 is missing explanations for the abbreviated labels. In general it would be more effective to use labels such as IndInfl than the labels you're using. I understand your space constraints, but the current format really challenges the reader to make sense of the tables and figures.
  • Although the paper is competently written from academic and grammatical standpoints, I'd suggest break some paragraphs up into two. For example, lines 274 to 288 would be easier to read if split into two paragraphs: one on correlation and one on regression results. Check this throughout.
  • Discussion: You state: "It is worth noting that the profile of components used 319 obtained in this group differs significantly from the one obtained in the group of city mayors [41], 320 which can be indicative of the fact that the office governance style depends on the rural or urban 321 character of the commune." This point needs more elaboration! If you have such comparative data to draw upon please expand this. It seems quite relevant and useful to extend our understanding of RURAL leadership processes.
  • The discussion/conclusion MUST come back to establishing the meaning and implications of this research for rural leadership for stakeholder engagement (social sustainability) in other societies. Readers will have no inherent interest in Poland. Please provide a bridge that links your findings to rural leadership for sustainability in other places (e.g., Egypt, Thailand, Italy).

Reviewer 2 Report

The manuscript is very interesting and discusses important issues. The article is remarkable and generally well written.

I have some remarks t that in my opinion may improve the paper:

  1. In the article, very little attention was devoted to linking analyzed management styles and sustainable development issues. How does management style affect the achievement of sustainable development goals? Is it possible to indicate based on research which style is the best for these purposes? Please develop this issue in the paper, in the literature review and discussion sections.
  2. Materials and Method:

How were calculated the three measures of results (employee satisfaction, effectiveness, extra effort)? According to mayors answers? How do the Authors understand and define effectiveness?

  1. In the section Results:

There is no comment on the results in Table 1. What should be noted? Which result and why do Authors find interesting?

For the Readers it would be helpful to indicate in the Table 1 which component applies to which leadership style (transactional and transformational).

The paragraph: the collected data was compared to the distributions obtained in numerous studies conducted by the authors of the questionnaire and the resulting values were categorised as low, medium or high based on percentile levels in the control group  requires a detailed description.  

The results may support t or explain the hypothesis rather than confirm it.

Results presented in the table 5 need to be discussed further.

  1. Discussion

“these behaviours are characteristic of transformational leadership, whose elements are not commonly used by commune mayors in the Greater Poland Province. This limits the possibilities of implementing the postulates and realising the goals of sustainable development” – Why? Please discuss and support this opinion.

  1. Conclusions:

Please identify clearly any implications for research, practice and/or society? How can the research be used in practice (economic and commercial impact), in teaching, to influence public policy, in research (contributing to the body of knowledge)? What is the impact upon society (influencing public attitudes, affecting quality of life)?

  1. References:

There is almost half of the references list in polish. For non-Polish language readers, this impedes accessing this literature. Also, a small percentage of the most current (up to date, after 2015) literature is cited.
